# Incidence, Etiology, and Risk Factors of Clinical Mastitis in Dairy Cows under Semi-Tropical Circumstances in Chattogram, Bangladesh

**DOI:** 10.3390/ani11082255

**Published:** 2021-07-30

**Authors:** Shuvo Singha, Gerrit Koop, Ylva Persson, Delower Hossain, Lauren Scanlon, Marjolein Derks, Md. Ahasanul Hoque, Md. Mizanur Rahman

**Affiliations:** 1Department of Medicine and Surgery, Chattogram Veterinary and Animal Sciences University, Chattogram 4225, Bangladesh; md.hoque@my.jcu.edu.au (M.A.H.); mizanuhcp@cvasu.ac.bd (M.M.R.); 2Udder Health Bangladesh, Chattogram Veterinary and Animal Sciences University, Chattogram 4225, Bangladesh; g.koop@uu.nl (G.K.); ylva.persson@sva.se (Y.P.); delowervet@sau.edu.bd (D.H.); Lauren.Scanlon@tufts.edu (L.S.); marjolein.derks@wur.nl (M.D.); 3Department of Population Health Sciences, Faculty of Veterinary Medicine, Utrecht University, Yalelaan 7, 3584 CL Utrecht, The Netherlands; 4National Veterinary Institute, 751 89 Uppsala, Sweden; 5Department of Medicine and Public Health, Sher-e-Bangla Agricultural University, Sher-e-Bangla Nagar, Dhaka 1207, Bangladesh; 6Cummings School of Veterinary Medicine, Tufts University, 200 Westboro Road, North Grafton, MA 01536, USA; 7Farm Technology Group, Wageningen University and Research, Droevendaalsesteeg 1, 6708 PB Wageningen, The Netherlands

**Keywords:** incidence rate, pathogens, antimicrobial resistance, clinical mastitis, MALDI-TOF

## Abstract

**Simple Summary:**

Bovine clinical mastitis is an inflammatory disease of the mammary gland associated with visual changes in the milk and/or the udder. We show that the incidence of clinical mastitis in commercial dairy farms in Bangladesh is high but with large variation between farms. Streptococci and non-aureus Staphylococci were the most frequently isolated bacteria from quarter milk samples. *Staphylococcus aureus* and non-aureus Staphylococci were often resistant against penicillin and oxacillin. This work suggests an urgent need for improved udder health management and specifically a more prudent use of antimicrobial agents following a treatment protocol.

**Abstract:**

Clinical mastitis (CM) is an important production disease in dairy cows, but much of the knowledge required to effectively control CM is lacking, specifically in low-income countries where most farms are small and have specific dairy management, such as regular udder cleaning and practicing hand milking. Therefore, we conducted a 6-month-long cohort study to (a) estimate the incidence rate of clinical mastitis (IRCM) at the cow and quarter level, (b) identify risk factors for the occurrence of CM, (c) describe the etiology of CM, and (d) quantify antimicrobial susceptibility (AMS) against commonly used antimicrobial agents in *S. aureus* and non-aureus *Staphylococcus* spp. (NAS) in dairy farms in the Chattogram region of Bangladesh. On 24 farms, all cows were monitored for CM during a 6-month period. Cases of CM were identified by trained farmers and milk samples were collected aseptically before administering any antimicrobial therapy. In total, 1383 lactating cows were enrolled, which totaled 446 cow-years at risk. During the study period, 196 new cases of CM occurred, resulting in an estimated crude IRCM of 43.9 cases per 100 cow-years, though this varied substantially between farms. Among the tested CM quarter samples, Streptococci (22.9%) followed by non-aureus staphylococci (20.3%) were the most frequently isolated pathogens and resistance of *S. aureus* and NAS against penicillin (2 out of 3 and 27 out of 39 isolates, respectively) and oxacillin (2 out of 3 and 38 out of 39 isolates, respectively) was common. The IRCM was associated with a high milk yield, 28 to 90 days in milk, and a higher body condition score. Our results show that there is substantial room for udder health improvement on most farms.

## 1. Introduction

The dairy sector in Bangladesh is progressively emerging, and a major part of the national milk supply is produced by cross-bred cows [1,2]. In developing countries like Bangladesh, an emphasis has been put on high milk production to alleviate poverty and to meet the daily requirement of milk of 250 mL/person [3]. The present national production is 69% of the quantity of milk needed to be self-sufficient according to recent reports [3,4]. For this purpose, across the country, indigenous cows have been replaced by cross-bred cows (Holstein Friesian × Indigenous and Holstein Friesian × Sahiwal × indigenous) [5] using a national AI program since 1959 [6,7]. However, these cows are more susceptible to production diseases like mastitis [8,9].

Globally, clinical mastitis (CM) is an important production disease in the dairy industry and has a great economic impact because of reduced milk yield, milk quality deterioration, treatment costs, involuntary culling, death, increased risk of antimicrobial resistance (AMR), and reduced animal welfare [10,11]. This impact might be even bigger in developing countries, where the number of cows per household is lower and milk is sold to consumers either directly or via local milk collection centers, creating a direct impact on food safety.

Many studies have previously been conducted to determine the incidence rate of CM (IRCM) in various countries across the world, e.g., Canada [12,13], Brazil [14], Tanzania [15], Ethiopia [16], Thailand [17], and several European countries [18,19]. The estimates of the IRCM have been widely variable between studies and regions. Although there are some published reports on bovine CM in Bangladesh [20,21,22], these only estimated the prevalence of CM, as no prospective studies (cohort studies) were performed. Additionally, the etiology of CM varies between different geographical locations and farm types. Among the CM-causing microorganisms, *Staphylococcus* (*S.*) *aureus*, non-aureus staphylococci (NAS), *Streptococcus* (*Strep.*) *agalactiae*, and *Mycoplasma* spp. are considered contagious, whereas *Strep. dysgalactiae, Strep. uberis*, *Escherichia* (*E.*) *coli*, and *Klebsiella* spp. are considered major environmental organisms. Species of NAS as well as *Strep. uberis* and *Strep. dysgalactiae* have been found to be either contagious or environmental [18,23,24,25]. As far as the authors are aware, bacterial agent results from previous studies in Bangladesh have only been reported from subclinical mastitis (SCM) [26,27].

In Bangladesh, about 80% of cattle farmers employ backyard household farming tethered in an enclosed area, while only 13% of farmers have an intensive or semi-intensive cattle housing system and 60% of farms have a concrete floor. Cows are generally milked by hand. Usually, no bedding materials are used. The cows are mostly fed rice straw as a source of roughage and only 40% of the farmers have land to grow fodder for their cows [28]. Most farmers practice stall feeding and only 37% of the farmers allow pasture grazing of the cows. This scarcity of feed combined with insufficient knowledge of cattle husbandry are important determinants of animal health in this setting and thus are major constraints of the dairy industry in Bangladesh [29].

Cow-specific risk factors in association with CM have been insufficiently studied in dairy cows in Bangladesh. Parity, breed, lactation stage, and previous history of CM may contribute to the likelihood of developing CM [30,31]. Dirty udders and hyperkeratosis of teats have also been associated with an increased risk of CM [32]. Knowing cow-specific factors that explain the variation in CM burden between cows may identify how extra care can be taken for high-risk animals.

Antimicrobial susceptibility (AMS) patterns against CM pathogens have been reported in various places around the world [33,34,35], and vary largely by region. As there are also no national guidelines for the use of antimicrobials in the treatment of CM cases in Bangladesh, knowledge of AMS of pathogens causing CM is needed to develop evidence-based treatment guidelines.

This study is the first to prospectively study CM in Bangladesh, allowing us to effectively estimate the IRCM. Therefore, the aims of the present study were to (a) estimate the IRCM at the cow and quarter level, (b) identify risk factors for the occurrence of CM, (c) describe the etiology of CM, and (d) quantify AMS against commonly used antimicrobial agents in *S. aureus* and NAS in dairy farms in the Chattogram region of Bangladesh.

## 2. Materials and Methods

### 2.1. Herd Selection

The Chattogram district of Bangladesh is located in the south-eastern part of the country at 22°0′45″ N and 92°6′1″ E. This district is subdivided into 18 administrative areas called upazillas or thanas. A comprehensive list of 102 dairy farms, each consisting of at least 2 lactating cows, within the Chattogram district was created by the Udder Health Bangladesh (UHB) team at Chattogram Veterinary and Animal Sciences University (CVASU) with the support of the upazilla livestock office and veterinary hospital. Initially, 2–3 villages or wards per upazilla or thana were chosen based on the highest cattle population density in the Chattogram district [36]. From there, a cohort of 24 farms (Figure 1) (representing 9 upazillas or thanas, each contributing between 1 and 9 farms) consisting of 1383 lactating cows were selected based on the willingness of the farmer to participate in a 6-month prospective cohort study (May to October 2018). The selected 24 dairy farms in this study were commercial farms, which we defined as farms rearing ≥9 cross-breed cows, adopting stall-feeding, which sell most of their milk. Commercial farms were chosen as this is the farm type we expect to become most important in the future. The study was approved and performed in line with the guidelines of the animal experimentation ethics committee (AEEC) of CVASU and written consent was taken from each farmer to participate in the study.

### 2.2. Data Collection

A questionnaire was prepared and pretested with the selected 24 farmers one month prior to the main study. The questionnaire was administered either to the farm owner, manager, or responsible farm employee through face-to-face interviews and data were collected by performing observations on-farm. Baseline data (both farm and animal level) were collected from each selected farm at the start of the study, including the location of the farm, farmer’s demography, farm composition (number of animals), housing system (face-in: head to head arrangement; and face-out: tail to tail arrangement of cows in the shed), source of animals (own source/purchase), presence of foot bath, farm environment cleanliness (yes/no) (if the pathways, feeding area and gutters were cleaned and disinfected and were dry at the time of observation), floor cleanliness (yes/no) (if lactating cow shed floor was cleaned and disinfected and were dry at the time of observation), milking system (hand milking and/or machine milking), calf sucking, number of milkers, and milking experiences of the milkman. For all lactating cows in each herd, the following data were collected: animal identity number; age; parity; body condition score (BCS), which was measured on a scale of 1 to 5 [37] and categorized (low: less than 3.0; moderate: 3 to 3.25 and high: 3.26 or more) [38]; milk yield (liters/day); and days in milk (DIM). A bidirectional communication system using a phone call or face-to-face meeting was developed between the UHB team and the farmers to execute farm monitoring. All the farms were monitored by contacting trained farmers on a weekly basis and farmers informed the UHB team if they detected any incidence of CM. The aforementioned individual animal variables were recorded for each new cow that entered into the lactating herd at regular monthly visits performed by the same investigators. At the beginning of the study, a training program was organized for the farmers to understand and learn how to engage with the structured monitoring of the farms in recognizing CM cases, collecting samples, and recording required information. The information in the record keeping sheet was thoroughly discussed to gather necessary data and a handout on clinical signs of CM with images were provided in Bengali (mother tongue of the farmers) to each farmer for easy recognition of different categories of CM. After training, the farmers were provided with sterilized falcon tubes, cotton, antiseptics (Hexisol^®^- Clorhexidine gluconate 0.5% *w*/*w* in 70% isopropanol, ACI Limited, Bangladesh), a marker pen, tape, a simple sample collection procedure flow chart in Bengali, a schedule for the farm visits to record dynamic data (entry date of new lactating cows in the herd, exit date of dry, sold, and dead cows), and recording sheets for information on cases of CM. Climatic conditions (temperature and relative humidity) were recorded using a digital hygrometer (PRESTO^®^ digital hygrometer, Taiwan) at monthly intervals. At each visit, observations of the temperature (°C) and relative humidity (%) were taken in the shed commonly used for milking and housing between 9 and 11 a.m. Data were recorded by face-to-face interviews following the field visit scheduled at regular intervals.

### 2.3. Case Definition

Clinical mastitis was classified according to the criteria described by Pinzón-Sánchez and Ruegg [39]: grade-1 (mild), when only the milk was abnormal; grade-2 (moderate), when abnormal milk was accompanied by swelling, and/or redness, and/or pain of the mammary gland; and grade-3 (severe), when the cow exhibited systemic signs of illness, such as depression, anorexia, dehydration, or fever, in addition to abnormalities in the milk and udder. Whenever any farmer identified a cow with CM, the farmer was requested to record the clinical signs in a pre-structured record keeping sheet, collect quarter milk samples aseptically before giving any drug, and store these samples in a freezer (−10 to −15 °C). The farmer of the affected cow’s farm was also requested to inform the UHB team immediately, which then visited the farm within 1–5 days to collect the samples and record the following information: cow identification number, BCS, parity, milk yield at the last milking before detection of CM, and DIM. Once a cow had a case of CM affecting one or multiple quarters, the cow exited the at-risk population and was not followed up for recurrence of CM.

### 2.4. Milk Sampling and Laboratory Analysis

Milk samples (2–5 mL) were separately aseptically collected from one or multiple affected quarters of a cow by trained farmers on site using 15 mL sterile falcon tubes with a unique identification number. Before sample collection, the teat-ends were cleaned using a sterile cotton ball soaked in 70% isopropanol to ensure aseptic collection according to the National Mastitis Council (NMC) guidelines [25]. The frozen milk samples were collected on farm and transferred within 3 h to the laboratory of CVASU in insulated ice boxes and stored at −20 °C before the samples were subjected to bacteriological culture.

#### 2.4.1. Bacteriological Culture

Milk samples were subjected to bacteriological culture in accordance with the NMC guidelines [25], with some modifications as noted below. Quarter milk samples were inoculated (10 µL) on bovine blood agar (BBA). The presence of at least three morphologically similar colonies was defined as positive growth. Gram-positive bacteria were identified based on the growth characteristics on BBA, then mannitol salt agar (MSA) followed by the catalase test (3% H_2_O_2_), coagulase test (using horse plasma), and gram staining. If the colonies on BBA had *β* hemolysis, a positive catalase test, yellow color formation on MSA, and a positive result in the tube coagulase test, then they were considered *S. aureus*. Those with no hemolysis (white, yellow, or golden colonies) on BBA, a positive catalase test, yellow or pink color formation in MSA, and a negative coagulase test were considered NAS. Minute transparent colonies on BBA with/without *β* hemolysis, a negative catalase test, and violet-colored short- or long-chain cocci in Gram stain were considered streptococci. Large grayish colonies with *β* hemolysis with large violet bacilli in Gram staining were considered *Bacillus* spp. Gram-negative bacteria were presumed based on small opaque colonies with fecal odor on BBA and growth on MacConkey agar. They were identified based on the morphology on MacConkey agar (MAC), then Eosin methylene blue agar (EMB) followed by IMViC tests (Indole test, Methyl Red test, Voges-Proskauer test, Citrate test), oxidase, and motility tests. *Escherichia coli* was presumed on findings of small opaque colonies with fecal odor in BBA, pink colonies in MacConkey agar, metallic sheen on EMB agar, negative motility test, negative citrate test, negative oxidase test, and positive indole and MR tests. Isolates were preserved at −80 °C using Brain Heart Infusion Broth and 50% buffered glycerol for future use. All the bacterial culture media and reagents were purchased from Oxoid Ltd., Basingstoke, United Kingdom. If more than 2 morphologically different colony types were found in a culture, the sample was considered contaminated. If two morphologically different colony types were found, both were identified based on the description above. For statistical analyses, the finding of a *Bacillus* spp. or non-specified Gram-negative bacteria (NGN) with other species was ignored and only the other pathogens cultured were reported. When two morphologically different NAS colony types or two *Streptococcus* spp. colony types were found in the same sample, this was reported as NAS or *Streptococcus* spp., respectively. When NAS were found, in combination with *S. aureus*, only the *S. aureus* was reported [40] as the latter species is more virulent and therefore more likely to have caused the CM than NAS as *S. aureus* is generally seen as a major pathogen and NAS as minor pathogens.

#### 2.4.2. MALDI-TOF Analysis, Real-Time PCR, and MIC

All presumptive *S. aureus* isolates and the subset of 12 randomly selected presumptive NAS isolates were transferred for confirmation considering the time and funding limitation to the reference mastitis laboratory, National Veterinary Institute (SVA), Uppsala, Sweden for MALDI-TOF using the MALDI Biotyper 3.0 (Bruker Daltonics GmbH, Bremen, Germany) analysis to confirm the accuracy of the phenotypical identification. The *S. aureus* isolates were subjected to real-time PCR (RT-PCR) to test for methicillin resistance. DNA extraction was performed by heating 1 µL of colony material mixed in 200 µL of Tris-EDTA buffer (pH 7.5) for 15 min at 95 °C. Centrifugation was performed for 3 min at 21,000× *g* relative centrifugation force (rcf) and the resulting supernatant was collected for further analyses. Four genes, *mecA* and *mecC*, *lukSF-PVL* (Panton-Valentine Leukocidin), and *nuc* (*S. aureus* specific marker), were selected, and real-time quadriplex PCR was performed in an Applied Biosystems 7500 Fast Real-Time PCR System (Foster City, CA, USA) using Quanta PerfeCTa Multiplex qPCR SuperMix (Quanta Biosciences, Beverly, MA, USA) according to the protocol described by Pichon et al. [41] and McDonald et al. [42]. 

A total of 44 isolates comprising 40 NAS (including 9 NAS isolates identified in mixed culture) and 4 *S. aureus* were subjected to MIC determination in broth micro dilution method using antimicrobial agent-coated microtitre plates (VetMIC CLIN staf/strept panel, SVA, Uppsala, Sweden). A total of 12 antibiotics (Penicillin, Cefalotin, Oxacillin, Cefoxitin, Enrofloxacin, Fusidic acid, Erythromycin, Clindamycin, Gentamycin, Nitrofurantoin, Tetracycline, and Trimethoprim/sulfamethoxazol) coated at different concentration provided by the manufacturer were used for evaluation during MIC determination. To check the validity of the MIC panel, *S. aureus* CCUG 15,915 which is an analogue to *S. aureus* ATCC 29,213, was used as a control strain and was tested in parallel with each of the batches. MIC values were interpreted based on MIC breakpoints available in the Clinical and Laboratory Standards Institute (CLSI, https://clsi.org/ (accessed on 25 November 2019) or European Committee on Antimicrobial Susceptibility Testing (EUCAST, http://www.eucast.org/ (accessed on 4 November 2019) or Swedres-Svarm (Swedish Veterinary Antimicrobial Resistance Monitoring) 2018 databases.

### 2.5. Statistical Analysis

The data obtained was entered into Microsoft Excel 2016 (MicrosoftCorp., Redmond, WA, USA) and data cleaning, coding, and integrity checking were performed before exporting to STATA IC-13 (Stata Corp. College Station, TX, USA) for the epidemiological analysis. R, version 3.5.3 (R Core Team, 2019) was used for the graphs.

#### 2.5.1. Descriptive Statistics

The IRCM at animal level was calculated as the number of animals that had at least one case of CM divided by the total animal-time at risk, and the results were scaled into 100 cow-years. Similarly, the IRCM at the quarter level was calculated as the number of quarters with CM divided by the total quarter-time at risk, and the results were scaled into 100 quarters-year. The IRCM at the animal and at the quarter levels was also calculated separately for each grade of mastitis as described above for each of the 24 farms. From six subsequent monthly observations on temperature and relative humidity, the average value at each farm was calculated. The presented 95% confidence intervals were calculated by applying the binomial exact method.

#### 2.5.2. Identification of Risk Factors for CM

Categorical variables were re-categorized as follows: BCS was categorized into three classes: low (≤3), moderate (3.25), and high (≥3.5). Parity was separated into five classes: 1, 2, 3, 4, and 5 or more. Based on the onset of CM, DIM was divided into four classes of the lactation stages: 27 or less, 28 to 90, 91 to 185, and ≥186. Milk yield was divided into four classes: ≤10 L/d, 10.1 to 13.0, 13.1 to 17, and ≥17.1 L/d following Bhat et al. [5]. A chi-squared test followed by a univariable random effect logistic regression model taking farm ID as a random effect was applied to evaluate the association between the binary response variable CM (yes or no) and the explanatory variables. Records with missing values were excluded from these analyses. The results of the univariable random effects logistic regression models were presented as subject-specific odds ratios to show the effect of the risk factors for an individual farm rather than averaged across farms. The multivariable random effect logistic regression model was constructed using the factors (milk yield, BCS, and DIM) that were significant at *p* < 0.2 in the univariable analysis. An ordinal multivariable random effect model was fitted with the variable ignoring the farm as the clustering variable. The same model was fitted again, including the farm as the cluster variable. The random effect model was then used to identify farm-level clustering. In the ordinary logistic regression model, a change (≥10%) between likelihood-based (model-based) standard error and robust (residual-based) standard error represented a clustered data set with the “farm ID” as the cluster variable. The model was fitted using a backward-elimination procedure of exposure variables by fitting the full model and the reduced model to a 20% level of significance. The relative difference of LRT (likelihood ratio test) based quadrature points (>0.01) at the 8th number cut-point confirmed that the final model fitted well. Confounding was checked for by adding or removing a variable from the model. The presence of multi-collinearity among the variables was tested by estimating the variance inflation factor. Finally, the Wald test significant variables found significant (*p* < 0.2) were considered as animal-level risk factors for CM and were expressed as an odds ratio (OR), 95% confidence interval, and *p*-value.

## 3. Results

### 3.1. Herd Characteristics

Farmers/managers/responsible persons (*n* = 24) had diverse educational qualifications ranging from secondary school or less (*n* = 8) to graduation or higher (Diploma, DVM, MBA, BBA, and PhD) (*n* = 16). Farmers’ perception about CM and antibiotics usage has been annexed as a Appendix A. Selected farms had a herd size ranging from 26 to 353, of which the numbers of lactating cows varied from 9 to 235. In the selected farms, 1383 cows with 5509 active quarters were followed up for a 6-month period, with a total of 446 cow-years at risk and 1907 quarter-years at risk, respectively. In 19 of the farms (*n* = 24), farmers allowed entry of new animals purchased from other sources. Farm environments were cleaned 1–4 times per day on 19 of the farms, but 5 farms were cleaned on a weekly basis or after an even longer period. Disinfectant foot baths were present at the entry point of 3 of the 24 farms. Farmers from 18 farms cleaned their shed floor 4 to 6 times a day, 5 farmers cleaned 3 times a day, and the one other farm cleaned the floor 2 times a day. The average temperature and relative humidity within farms ranged between 27.2 and 39.8 °C and 62% and 82.2%, respectively.

### 3.2. Incidence Rate of Clinical Mastitis at Different Levels

At the animal level, the IRCM was 43.9 cases (196 cases/446.25 cow years at risk) per 100 cow-years. The IRCM values of grade-1, grade-2, and grade-3 are given in Table 1 at the cow level and at the quarter level. Between farms, the cow-level IRCM ranged from 0 (0 cases/14.1 cow years at risk) (95% CI: 0–23.2) to 103 (15 cases/14.49 cow years at risk) (95% CI: 62.4 to 171.7) cases per 100 cow-years (Figure 2). No cases of CM were detected on 2 of the farms during the 6-month monitoring.

### 3.3. Risk Factor Analysis

Table 2 gives the descriptive information and univariable associations between the IRCM and four cow-level variables: BCS, parity, daily milk yield, and DIM. A positive association was also observed in the multivariable logistic regression (mean VIF-1.45) between CM and BCS, between CM and high milk yield, and CM was most likely to happen between 28 and 90 days of DIM (Table 3).

### 3.4. Bacteriological Culture Results of Milk Samples from Clinical Mastitis

A total of 222 affected quarters in 196 cows and 22 farms were reported for CM. During the follow-up period, 153 collected quarter milk samples (68.9%) were submitted by farmers. The rest of the quarter samples were not received or discarded because some farmers mistakenly collected samples after antimicrobial treatment (*n* = 3), had not used a freezer immediately for sample storage (*n* = 5), or did not submit samples (*n* = 61). Two of the farms had no incidence of CM during the study period and one of the farms had a case of CM but did not submit any milk samples. Therefore, milk samples from 21 farms were included in the bacteriological analysis. Of the 153 quarter milk samples from CM cases, 18.3% of the samples were culture negative, and 1.3% of the samples were contaminated. The most frequently isolated organisms were *Streptococcus* spp. (22.9%) followed by NAS, *Bacillus* spp., NGN, *E. coli*, and *S. aureus* (Table 4).

### 3.5. MALDI-TOF Testing of Selected Isolates

Among the 16 isolates tested by MALDI-TOF, the 4 presumptive *S. aureus* isolates were confirmed as *S. aureus,* and the 12 presumptive NAS isolates were identified as NAS (*S. sciuri* (*n* = 5), *S. haemolyticus* (*n* = 4), *S. arlettae* (*n* = 2), and *S. cohnii* (*n* = 1)).

### 3.6. MIC Determination and Real-Time PCR

In total, 39 isolates of NAS and 3 isolates of *S. aureus* underwent antimicrobial susceptibility testing through determination of the MIC values. In the MIC analysis, two of the isolates (one NAS and one *S. aureus*) were missing as they did not grow in the sub-culture during MIC testing. Among the *S. aureus* isolates, one was detected as methicillin-resistant *S. aureus* (MRSA) according to the RT-PCR with the presence of *nuc* and *mecA* genes and the other three were identified as methicillin-sensitive *S. aureus.* The distribution of MIC against 42 isolates is given in Table 5. Following MIC, the NAS isolates were found resistant to erythromycin (66.7%), penicillin (69.1%), and oxacillin (97.4%).

## 4. Discussion

In the present study, we aimed to estimate the IRCM at cow and quarter levels and we compared how these estimates varied between herds. Results revealed that a wide variation exists between farms. The predominant species causing CM were *Streptococcus* spp. and NAS, and both *S. aureus* and NAS had high levels of resistance against penicillin and oxacillin. The characteristics of the cows and selected farms in this study were representative of most of the dairy farms located in other parts of Bangladesh [1,9,43] and to some extent to farms in neighboring and other developing countries [16,17,44,45] and thus the findings of this prospective study might be relevant for the aforementioned farming situations.

### 4.1. Incidence Rate of Clinical Mastitis at Different Levels

In the present study, the IRCM was fairly high at the animal and quarter levels, consistent with the findings of numerous studies reported in other parts of the world who also reported the IRCM at cow or quarter levels [15,18]. The cow-level IRCM was higher than those in Canada and the Netherlands [13,19] but lower than what was previously reported in Brazil [14]. The available IRCM values reported in previous studies in various countries are given in Table 6. However, we could not compare the estimate in Bangladesh or neighboring countries due to a lack of data to support such comparisons. The high IRCM could be associated with inadequate or poor farm management practices [43] and stress conditions that frequently exist in the intensive farming system used in Bangladesh [46]. The IRCM was significantly higher for grade-2 at both animal and quarter levels compared to grade-1 and -3 in the present study, which does not correspond to two studies from Brazil and Belgium [14,47], who reported a predominance of grade-1 CM. The reason behind this difference might be misclassification. Although all farmers in the study were trained, some farmers might not have recognized grade-1 mastitis as clinical cases. Thus, better CM monitoring could be achieved by training the farmers to detect grade-1 mastitis more accurately to better estimate the overall CM burden in farms, which may have been underestimated in the current study. The relatively high grade-2 IRCM compared to previous studies may also reflect a true difference in CM severity between countries, resulting from factors linked to the different geographical location or possible protective factors, such as immunity of the cows [48,49].

There was a large variation of IRCM between farms. A high variation in IRCM between farms was also found in other studies [12,13]. Among the farms included in this study, there was a wider difference in herd-level factors, such as farmers’ education, herd size, cleaning and disinfection of the milking shed floor, and the farm environment. Furthermore, this is also likely the result of variation between farms in management practices, such as dry cow therapy, management of cows in the early and late dry period, and provision of separate calving areas [50]. These factors together could be associated with the transmission within a herd resulting in a higher IRCM but was not further explored in this study. No cases of clinical mastitis were detected at two farms in the present study, which could have been due to the maintenance of better management practices.

From previous farm records, these farms rarely experienced observable cases of clinical form of mastitis prior to our study. One farm uses sand as the bedding material, regularly feeds silage, cows are not tied, and the monthly bulk milk SCC is always <200,000 cells/mL of milk. The other farm owner is very experienced, and monitors the animals closely. 

### 4.2. Factors Associated with Clinical Mastitis

A higher milk yield was identified as a risk factor for CM, which is supported by several previous studies [14,51]. The teat canal diameter and stretchability are correlated with milk yield and thus are greater in high-yielding cows [52]. The teat canal usually remains open for a comparatively longer period in cows yielding larger volumes of milk, which may lead to an increased risk of mastitis [53] mainly by environmental mastitis pathogens, such as *E. coli* [54]. However, the genetic susceptibility of individual cows can also have a major influence on CM occurrence and may be correlated with other described factors [8,55]. Additionally, milk yield is higher in cross-breed cows than in indigenous cows in Bangladesh. The prevalent climatic conditions of tropical countries (high temperature and humidity) and intensive farming systems could impose heat stress on the higher-yielding cross-breed cows more than local-breed cows [56], possibly impairing the immune function of the cows and resulting in bacterial invasion and multiplication [49].

Cows with higher BCS were more susceptible to CM, which is a surprising result and is not consistent with many previous reports [15]. In a previous report, a lower BCS was associated with the occurrence of CM (OR 7.3) [22]. Alternatively, cows with higher BCS are more susceptible to other diseases, such as lameness and milk fever [57], which may result in an immuno-suppressed state, potentially explaining the association with CM [58].

An OR significantly > 1 was found for 28 to 90 days in lactation compared to >185 days in lactation (Table 3) and days in milk therefore seems to be a significant risk factor for CM in this study, which is in accordance with previous studies [31,59]. The peak of the milk production generally occurs around the ninth week after calving [60], which may explain the higher incidence in this period. However, the difference in IRCM between the first category (51.4 cases/100 cows/y) and the second category (56.2 cases/100 cows/y) was proportionally smaller than the difference in OR between these categories (1.4 and 2.1 respectively), suggesting that the absolute number of cases was diluted in a larger number of cow-years at risk for the category 28–90 days in milk, which makes sense, as this category spans a larger number of days than the lower category. As the category levels of the days in milk variable do not represent an equal follow-up time, the multivariable method gives an overestimation of the odds in the categories spanning a bigger time interval and should thus be interpreted cautiously. When, instead, looking at the IRCM (Table 2) this is taken into account, IRCM values of the early lactation categories (27 or less, and 28–90 days) were approximately similar and seemed higher than the categories in later lactation, although the difference in IRCM was not formally tested. Therefore, we conclude that mastitis incidence is probably higher in early lactation, but more data is needed. 

Farm-level factors, such as general cow cleanliness, farmer education, and herd size, can also influence IRCM. Although data related to these factors were collected, they were not analyzed as a 24-farm study is under-powered to detect the impact of such farm-level effects.

### 4.3. Pathogens Associated with Clinical Mastitis

*Streptococcus* spp. were the most prevalent pathogens (22.9%) followed by NAS (20.3%). Other than *S. aureus* and the selected subset of NAS, we did not identify the pathogens at the species level, which is a limitation of this study. However, among the identified subset of NAS isolates, we found *S. sciuri*, *S. haemolyticus*, and *S. cohnii*, which were previously described as environmental mastitis pathogens [61]. Both contagious and environmental mastitis causing staphylococci were present in the farms. However, transmission of the species of streptococci is variable and largely depends on the contagious or environmental nature and we did not know whether strep species were *S. uberis* (environmental) or *S. agalactiae* (contagious) or *S. dysgalactiae* (in between) because further species-level identification for streptococci was not performed. Low levels of *S. aureus* in the CM cases observed in our study might be because we only looked at new cases of mastitis, whereas *S. aureus* often causes chronic mastitis due to persistent IMI [62,63]. In order to present the incidence of new CM cases in this prospective study, we excluded the chronic mastitis cases from the susceptible population at risk. The most frequently found environmental pathogens were *Bacillus* spp. (19.0%) followed by *E. coli* (7.2%), although the high proportion of *Bacillus* spp. may have resulted from contamination [64]. The identified environmental pathogens’ source may be the farm floor or surroundings while the contagious pathogens are transmitted between cows by inappropriate milking practices [65]. Among the studied farms, some farms clean the farm premises less than once in a week, which might have played a significant role in contaminating the farm environment with the environmental mastitis-causing pathogens. The authors of [19] indicate farm hygiene and good milking practices are mandatory. However, mastitis pathogens could be homogenous; however, the pathogens vary widely globally according to the climate, environment, and management. We found a very low number of “*S. aureus*” in our study, which is less common than in other neighboring countries. Like in India, in a particular study [33], they found *S. aureus* to be most common pathogen in CM whereas in Pakistan Gram-negative bacteria were the most common [66]. So, there might be an association with the differences in the manageable farm-level factors, such as the milking system; calf sucking; shed, milker, and cow hygiene; and also the adoption of mastitis control management like teat dipping and dry cow management.

### 4.4. Antimicrobial Resistance

Among the three isolates tested, two *S. aureus* exceeded the epidemiological cut-off of the MIC value against penicillin (0.25 µg/mL) and oxacillin (0.5 µg/mL) and one isolate of these was positive for *nuc* and *mecA* genes and was thus genotypically confirmed as MRSA. However, drawing a conclusion on the prevalence of MRSA was not possible based on the identification of only one isolate. This finding indicates that more research is needed that includes more milk samples to estimate the prevalence of MRSA. Still our results indicate that public health safety measures are needed to prevent the transmission of MRSA through raw milk in the food value chain in this region. In addition, 69.1% of the NAS isolates also exceeded the cut-off for penicillin resistance (69.1%) and, surprisingly, 97.4% of NAS also exceeded the limit for oxacillin. Naccache et al. [67] previously reported about the capability of *S. epidermidis* to carry methicillin resistance. Furthermore, the oxacillin and penicillin resistance of NAS is also in line with Cheng et al. [34], who reported that NAS were 84% resistant against oxacillin followed by 62% against penicillin. The high level of oxacillin resistance in NAS isolates may have been overestimated, as EUCAST uses a higher threshold than the Swedres-Svarm 2018 threshold used by us. This may also partly explain why penicillin resistance was lower than the oxacillin resistance in these isolates. Poor internal and external biosecurity along with extensive and unjustified use of antimicrobials in the dairy farms in Bangladesh could have possibly caused high levels of AMR [26]. Nevertheless, cases of mastitis in Bangladesh are frequently treated with penicillin and other classes of antimicrobial agents, which could be associated with the high level of oxacillin resistance. More research to identify which antimicrobials are used on Bangladesh dairy farms and for what reason is needed to identify the actual causes of the high levels of oxacillin resistance and to identify possible interventions to reduce AMR. Due to time and funding constraints, only Staphylococci were included in the AMR part of this study. Further Bangladesh-focused research on AMR in other mastitis bacteria is therefore required, especially for Streptococci.

## 5. Conclusions

In this study, we found a high IRCM, which varied substantially between farms, suggesting there is room for improvement on most farms. The association of higher BCS, higher milk yield, and early lactation stage with IRCM shows that controlling CM may be challenging as it may conflict with the aim to improve production levels. On the other hand, CM also leads to milk yield reduction, so more research is needed to quantify whether preventive measures are cost effective. In the present study, both *S. aureus* and NAS displayed concerning levels of resistance against oxacillin and penicillin, which necessitates a better understanding of the drivers of AMR in dairy cows in Bangladesh.

## Figures and Tables

**Figure 1 animals-11-02255-f001:**
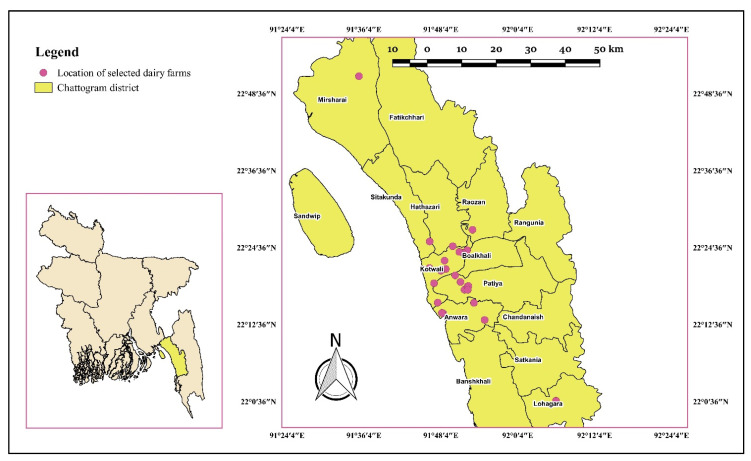
Location of the 24 commercial dairy farms located in 9 upazilla/thana under the Chattogram District in Bangladesh that were enrolled in the study.

**Figure 2 animals-11-02255-f002:**
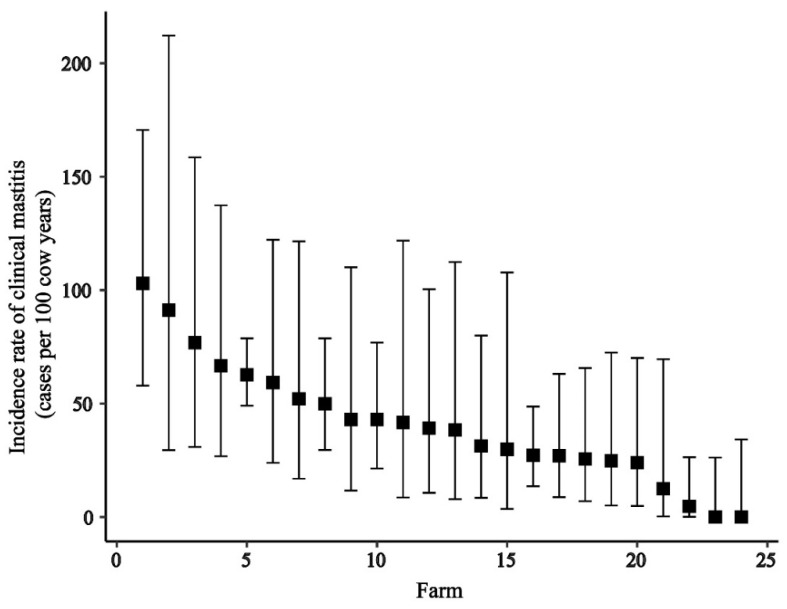
Farm-level incidence rate of clinical mastitis (cases per 100 cow-years) on 24 dairy farms in the Chattogram region in Bangladesh. The black box indicates the calculated cow-level incidence rate of clinical mastitis of each of the farms. The upper and lower limits indicate the 95% CI of the incidence rate.

**Table 1 animals-11-02255-t001:** Incidence rate of clinical mastitis in 24 commercial dairy farms in Chattogram, Bangladesh. The cow-level incidence rate (196 cases of clinical mastitis) was calculated in 1383 dairy cows and quarter-level incidence rate (222 cases of clinical mastitis) was calculated in 5509 quarters.

Level	Status	IRCM ^1^ per 100 Quarter-Years	95% CI	Level	Status	IRCM ^1^ per 100 Cows per Year	95% CI
Quarter level	Overall	11.6	10.2 to 13.3	Animal level	Overall	43.9	38.2 to 50.5
	Grade-1	2.8	2.1 to 3.6		Grade-1	10.5	7.9 to 14.0
	Grade-2	6.5	5.5 to 7.8		Grade-2	25.5	21.3 to 30.7
	Grade-3	2.4	1.8 to 3.2		Grade-3	7.8	5.6 to 10.9
Quarter position	Front right	10.8	8.3 to 14.2				
	Front left	14.2	11.1 to 18.0				
	Hind right	13.1	10.2 to 16.8				
	Hind left	8.5	6.3 to 11.6				
Quarter position	Front	12.5	10.4 to 15.0				
	Hind	10.8	8.9 to 13.1				

**^1^** IRCM: Incidence rate of clinical mastitis.

**Table 2 animals-11-02255-t002:** Univariable association ^1^ between the incidence rate of clinical mastitis and potential risk factors in 1383 dairy cows on 24 dairy farms in Chattogram, Bangladesh.

Factors	Categories	N ^2^	IRCM ^3^/100 Cows/y	95% CI (IRCM)	Odds Ratio	95% CI (OR)	*p*
Body condition score	≤3	427	34.9	26.5 to 45.9	Ref		
	3.25	488	45.7	36.0 to 58.0	1.3	0.9–1.9	0.2
	≥3.5	468	51.0	40.8 to 63.7	1.5	1.0–2.2	0.04
Parity	1	364	42.1	31.9 to 55.6	Ref		
	2	354	42.9	32.4 to 56.8	1.1	0.7–1.7	0.7
	3	279	37.5	26.9 to 52.2	1.1	0.7–1.7	0.7
	4	214	51.8	37.2 to 72.1	1.3	0.8–2.0	0.3
	≥5	172	51.4	35.3 to 75.0	1.1	0.7–1.9	0.7
Milk yield (L/d)	10 or less	462	30.8	23.0 to 41.3	Ref		
	10.1–13	266	38.6	27.6 to 54.0	1.4	0.9–2.3	0.1
	13.1–17	328	38.8	28.8 to 52.4	1.4	0.9–2.2	0.1
	>17	327	73.0	58.1 to 91.7	2.8	1.9–4.1	<0.001
Days in milk	27 or less	354	51.4	38.1 to 69.3	1.4	0.9–2.2	0.2
	28–90	392	56.2	44.9 to 70.4	2.1	1.3–3.2	0.001
	91–185	292	38.3	28.3 to 51.8	1.5	0.9–2.4	0.1
	>185	345	29.7	21.3 to 41.4	Ref		

^1^ Univariable logistic regression included the outcome (incidence of clinical mastitis) with either of the four variables (body condition score, parity, milk yield, and days in milk) using the farm identification number as a random effect; ^2^ N: Number of cows at each category; ^3^ IRCM: Incidence rate of clinical mastitis.

**Table 3 animals-11-02255-t003:** Multivariable association ^1^ between the incidence rate of clinical mastitis and potential risk factors in 1383 dairy cows on 24 dairy farms in Chattogram, Bangladesh.

Factors	Categories	N ^2^	Odds Ratio	95% CI (OR)	*p*
Body condition score	≤3	427	Ref		
	3.25	488	1.3	0.8 to 1.9	0.2
	≥3.5	468	1.4	1.0 to 2.2	0.08
Milk yield (L/d)	10 or less	462	Ref		
	10.1–13	266	1.3	0.8 to 2.1	0.3
	13.1–17	328	1.4	0.9 to 2.2	0.2
	>17	327	2.4	1.6 to 3.8	<0.001
Days in milk	27 or less	354	1.1	0.7 to 1.8	0.7
	28–90	392	1.6	1.0 to 2.7	0.03
	91–185	292	1.3	0.8 to 2.2	0.2
	>185	345	Ref		

^1^ Multivariable logistic regression included the outcome (incidence of clinical mastitis) with either of the three variables (body condition score, milk yield, and days in milk) using the farm identification number as a random effect; ^2^ N: Number of cows at each category.

**Table 4 animals-11-02255-t004:** Pathogen distribution from 153 clinical mastitis-affected quarter milk samples from 111 cows.

Name of Pathogens	*n* = 153	%
*Streptococcus* spp.	35	22.9
NAS ^1^	31	20.3
*Bacillus* spp.	29	19.0
NGN ^2^	13	8.5
*E. coli*	11	7.2
*Staphylococcus aureus*	4	2.6
Culture negative	28	18.3
Contaminated	2	1.3

^1^ Non-aureus *Staphylococcus* species.^2^ NGN = non-specified Gram-negative bacteria.

**Table 5 animals-11-02255-t005:** Resistance ^1^ (percent, 95% CI in brackets) and distribution (*n* isolates) of MIC for *Staphylococcus aureus* (*n* = 3) and non-aureus *Staphylococcus* species (NAS) (*n* = 39) from clinical mastitis in 34 dairy cows from 13 dairy farms in Chattogram, Bangladesh.

Test Agent	Species	Resistance ^1^ % (95% CI)	Distribution (*n* of MICs (µg/mL)											
			0.03	0.06	0.12	0.25	0.5	1	2	4	8	16	32	64
Penicillin	*S. aureus*	66.7 (9.4 to 99.2)				1		2						
	NAS ^2^	69.2 (52.4 to 83.0)	4	2	1	5	2	25						
Cefalotin	*S. aureus*	0 (0 to 70.8)						2		1				
	NAS ^2^	0 (0 to 9.0)						18	9	12				
Oxacillin + 2% NaCl	*S. aureus*	66.7 (9.4 to 99.2)					1	2						
	NAS^2^	97.4 (86.5 to 99.9)				1		38						
Cefoxitin	*S. aureus*	33.3 (0.8 to 90.6)				1	1				1			
	NAS ^2^	**-**				3	2	3	5	10	16			
Enrofloxacin	*S. aureus*	0 (0 to 70.8)				2	1							
	NAS ^2^	25.6 (13.0 to 42.1)				18	11	10						
Fusidic acid	*S. aureus*	33.3 (0.8 to 90.6)				2		1						
	NAS ^2^	0 (0 to 9.0)				19	7	13						
Erythromycin	*S. aureus*	0 (0 to 70.8)					3							
	NAS ^2^	66.7 (49.8 to 80.9)					13	4	22					
Clindamycin	*S. aureus*	0 (0 to 70.8)					3							
	NAS ^2^	43.6 (27.8 to 60.4)					22	7	10					
Gentamycin	*S. aureus*	66.7 (9.4 to 99.2)						1		2				
	NAS ^2^	30.8 (17.0 to 47.6)						18	9	12				
Nitrofurantoin	*S. aureus*	0 (0 to 70.8)										3		
	NAS ^2^	5.1 (0.6 to 17.3)										35	1	2
Tetracycline	*S. aureus*	33.3 (0.8 to 90.6)					1	1		1				
	NAS ^2^	43.6 (27.8 to 60.4)				11	7	3	1	17				
			Distribution (*n*) of MICs (mg/L)											
			0.25/4.75		0.5/9.5		1/19		2/38		4/76			
Trimethoprim + Sulfamethoxazole	*S. aureus*	0 (0 to 70.8)	2		1									
	NAS ^2^	0 (0 to 9.0)	13		5		5		4		12			

^1^ Resistance percent was calculated as the number of isolates identified as resistant according to the cut off divided by the total number of isolates tested for a particular type of bacteria (*S. aureus*/Non-aureus Staphylococci (NAS ^2^)); White fields denote the range of dilutions tested for each substance. Blue fields denote the dilutions were not available for the corresponding substance. In the MIC plate, there was a designed well for Trim-sulfa that contained the combination of 0.25 µg/mL Trimethoprim and 4.75 µg/mL Sulfamethoxazole. MIC of the above range is given as the concentration closest to the range. Bold vertical lines indicate the available epidemiological cut off values from EUCAST (https://eucast.org/ (accessed on 4 November 2019)) (Cefalothin, Nitrofurantoin), CLSI 2018 M-100 (S-28) (https://clsi.org/ (accessed on 25 November 2019)) (Penicillin) and Swedres-Svarm 2018 (Oxacillin, Cefoxitin, Clindamycin, Enrofloxacin, Erythromycin, Fusidic acid, Gentamycin, Tetracycline, Trimethoprim + Sulfamethoxazole). When no cut-off values were available, bacteria were not categorized as sensitive or resistant.

**Table 6 animals-11-02255-t006:** Comparison of estimates of the incidence rate of clinical mastitis between this study and other studies.

Incidence Rate (100 Cows/Year)	Quarter Level (100 Quarters/Year)	Number of Cows	Period of the Study	Country	Reference
23.3	-	12296	12 months	Canada	Olde Riekerink et al., 2008 [12]
23.7	-	5395	14 months	Canada	Levison et al., 2016 [13]
-	35.8	4374	15 months	Brazil	Tomazi et al., 2018 [14]
43.3	38.4	317	18 months	Tanzania	Kivaria et al., 2007 [15]
32.2	-	4947	12 months	Netherlands	Santman-Berends et al., 2015 [19]
41.6	-	810	12 months	UK	Bradley and Green, 2001 [24]
43.9	11.6	1383	6 months	Bangladesh	The present study

(-) Indicates data were not available in the cited reference.

## Data Availability

The data presented in this study are available within the article.

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
