# Peer review of "Incidence, Etiology, and Risk Factors of Clinical Mastitis in Dairy Cows under Semi-Tropical Circumstances in Chattogram, Bangladesh"

_animals, 2021, doi:10.3390/ani11082255_

Round 1

Reviewer 1 Report

This article tells us about clinical mastitis and risk factors of mastitis in Bangladesh. I find this text very well-written, and highly interesting in many respects, like food safety and AMR, and coping of the local people. Also you mention that most of the farms milk by hand, and I am particularly interested to see if it makes the situation and risk factors somewhat different. Here are my detailed comments:

Abstract: please open up other abbreviations than CM also.

M&M: Did you look at cow cleanliness? I find it tells more about the hygiene than floors, which can be cleaned for the farm visit.

Relative humidity you mean?

Results: 3.2. You mention no cases of clinical mastitis were detected at two farms. I am sorry, I do not understand what for example 0/14.1 mean? What are these farms that have no mastitis like? Do you trust that result?

Table 1. You do not need to repeat the 24 farms in the legend.

Front quarters appear to have more mastitis, which is very special. I would like to hear your comments on that in the discussion also.

3.3. ….CM was most likely to happen between 28 and 90 DIM (Table 3).

What about other risk factors? Farmer`s education, climate, milking system, herd size etc?

3.4. You do not need to repeat the results of Table 4 in the text.

In the Table 4. the text under the table is mysterious (starting with NAS-NAS etc.). I feel you try to explain the classification, and it should be done in M&M as it has been, no need to repeat here I believe.

Table 5. I see no vertical lines mentioned under the table?

Better would be: When no cut-off values were available, bacteria was not….

For Trim-sulfa what are the different MICs (0.25/4.75) for? I would leave Substance, Species and Resistance out here, but that is a matter of opinion.

Discussion:

Table 6. I find the text under the table not needed, since you have told this already in the introduction. Otherwise I like this table for putting things in perspective.

4.1. “,which does not correspond to (61,63) who….” sounds funny. I think you could use something like correspond to studies from Brazil etc (61, 63)….

4.2. BCS is indeed interesting. Did you make an interaction between DIM and BCS, because as you say it might affect cows at parturition?  

4.3. I am amazed, that all around the world mastitis bacteria appear to be the same, even though the climate, environment and management are so different. Could you have a comment on that?

Otherwise, I think this is the part of the article that needs more “flesh over the bones”. Now it seems you only repeat the results and say not much more. How does the lack of S. aureus relate to the milking systems used, for example?

4.4. Again, the funny …in line with (12) who…. A name would be nice.

You state that CM is frequently treated with penicillin and other antibiotics and that makes high oxacillin resistance not surprising. Why? Ain`t it a same situation in other parts of the world also?

Reviewer 2 Report

The study aimed to estimate the IRCM at cow and quarter level, identify risk factors for the occurrence of CM, and to describe the etiology of CM, estimate the AMS of in S. aureus and NAS in dairy farms in the Chattogram region of Bangladesh. The experimental design was a prospective study, during a 6-month period. The main finding of the study was to describe an IRCM of 44 cases per 100 cow-year and to observe a high level of antimicrobial resistance in S. aureus (only few isolates evaluated) and non-aureus staphylococci.

In general terms, the manuscript is well written, using a very good English language and style. The study brings good contribution about the epidemiology information of clinical mastitis in the selected region and also in similar conditions, which may interest readers with focus on mastitis epidemiology.

Minor review and specific comments are provided in the revised file.

Reviewer 3 Report

Shuvo Singha et al. explored iIncidence, etiology, and risk factors of clinical mastitis in crossbred cows under semi-tropical circumstances
Overall, the manuscript is interesting.
Udder diseases are the main cause of economic losses for breeders. Additionally, they cause pain to the animal. Therefore, the current topic is interesting and timely.

Author Response

Response by the authors to reviewers comments to manuscript animals-1222325

Dear respected reviewer,

Thank you very much for giving us the opportunity to revise our manuscript (Manuscript ID animals-1222325) entitled “Incidence, etiology, and risk factors of clinical mastitis in crossbred cows under semi-tropical circumstances”. This revision has been made following the valuable comments given by the four reviewers. We would like to express our sincere thanks to you for being one of the reviewers for your kind contributions to render an improved version of the manuscript from the original submission.

All responses are documented below, have “Authors:” as a preface to the response, and are in blue italic font.

Subject: animals-1222325 “Incidence, etiology, and risk factors of clinical mastitis in crossbred cows under semi-tropical circumstances”

3rd July 2021

Overall, the manuscript is interesting.

Udder diseases are the main cause of economic losses for breeders. Additionally, they cause pain to the animal. Therefore, the current topic is interesting and timely.

Authors (1): Thank you for your valuable opinions. The Language has been further checked to improve by all the expert co-authors in the manuscript

Reviewer 4 Report

Methodologically well structured article with very relevant results. The fact that there are large differences between farms in the incidence of clinical mastitis could be further discussed.

Round 2

Reviewer 1 Report

Thank you for considering the earlier comments. I think this is ready for publication now, and I am satisfied for your answers and reactions.

Author Response

Thank you so much

This manuscript is a resubmission of an earlier submission. The following is a list of the peer review reports and author responses from that submission.

Round 1

Reviewer 1 Report

The manuscript presented is a very interesting study on the incidence, etilogy and risk factors of clinical mastitis in cross-bred dairy cows in a district of Bangladesh. The authors presented a relevant amount of data of high interest for farmers, veterinaries and FBOs. The quality of the paper is high and I have only few concerns:

  • the line numbers are not present. This make difficult to give a proper review.
  • in the abstract there are some abbreviations without the corresponding extended form. I suggest to add them.
  • I suggest to take in consideration the possibility to cite the following papers: 1) Luca Grispoldi, Luca Massetti, Paola Sechi, Maria F. Iulietto, Margherita Ceccarelli, Musafiri Karama, Paul A. Popescu, Francesco Pandolfi, Beniamino T. Cenci-Goga. Characterization of enterotoxin-producing Staphylococcus aureus isolated from mastitic cows. J. Dairy Sci. 102:1059–1065. 2) Grispoldi, L., Karama, M., Ianni, F., La Mantia, A., Pucciarini, L., Camaioni, E., Sardella, R., Sechi, P., Natalini, B., Cenci-Goga, B.T. 2019. The relationship between Staphylococcus aureus and branched-chian amino acids content in composite cow milk. Animals, 9, 981.

Reviewer 2 Report

As it is a developing country with manual milking, the incidence of mastitis seemed to me to be underestimated. Diagnostic tests to confirm mastitis would need to be performed for the results to be valid: mug of black background test, California Mastitis Test (CMT), Somatic Cell Count.
Technical problems can interfere with microbiological diagnostic. In addition, the previous use of antimicrobials may have been responsible for the low incidence of mastitis found in this study.
Microbiological results need to be made available (CFU/ml), diagnostic of mastitis, animal, herd, farms.
Very short study period to determine the incidence of mastitis.

Reviewer 3 Report

Dear authors,

Thank you for an interesting and well written paper. I do only have a few comments below.

Simple summary:

In the last sentence, you say there is a need for a more prudent use of antibiotics. This is in my opinion a bit to far from what you are actually investigating. I do not see that you do records on treatments at all?  You are probably thinking of the high level of antimicrobial resistance, could you rephrase the sentence more in that way?

Abstract:

I do not agree in this statement “knowledge required to effectively control CM is significantly lacking” – actually I think we have a lot of knowledge – you can just take the 10-point plan. We had this knowledge for years! The hard thing is to implement this in our production systems. And this may be even more challenging in low-income countries, I agree with that.

I would like to have abbreviations defined in the abstract – especially AMS can have different meanings. First you use S.aureus and later on you use the full name. Please check the consistency.

“resistance against penicillin and oxacillin was common” – could you just add a few numbers? E.g. above XX.

Introduction:

“ Cow specific risk factors in association with CM have been insufficiently studied in dairy cows in Bangladesh. Parity, breed, lactation stage, and previous history of CM may contribute to the likelihood of developing CM [3,43].” - What makes you think that this may be much different in Bangladesh compared to other countries where lots of these studies are done? Could you add a sentence?

Materials and Methods:

Body condition score – you define here “low” as less than 3, in the table with results, you include 3 in your “low” category. For the “high” category I would use “above 3.25” or “3.5 to 5”. I guess you use the scale with 0.25 between each score?

The grouping you do on pathogens/when you exclude Bacillus when other pathogens occurred. Could you please explain why you do this and how many cases you had with this combined growth, where you excluded something? And did you do this before or after you decided if the sample was contaminated?

Results:

“In 19 of the farms (n=24)” what is this n=24 ?

How do you do you measure the grade (1,2,3) on cow level? If you had more quarters affected in the same cow and they always got the same score, I would like that information.

I think you collected a lot of information from the farms. You describe the general housing of cows in the introduction – could you give a bit more information about the included farms as descriptive results? Eventually on farm level, to look for a tendency in relation to figure 2.

Table 2 and 3: could you add to the table description that this was on cow-level? I guess it was?

Table 4: I do not understand the text below the table “NAS-NAS”?

Table 5: the table looks a bit confusing. Also, there are no bold vertical lines – please check the layout.

Discussion:

“Cows with higher BCS were more susceptible to CM, which is a surprising result and not consistent with previous reports [35].” – but this was also not the case for the multivariable model?

In general I think you have the material to provide a few more thoughts about your results. Regarding the pathogens, the no. of no-growth samples and the high incidence of CM – how was the hygiene actually in these farms, at milking? In the barn? the management? And regarding the AMR: how were mastitis treatments mainly done? You wrote penicillin, but length of treatment and administration route? Did farmers treat all CM or did they have any criteria?